# Perceived Barriers of Accessing Healthcare among Migrant Workers in Thailand during the Coronavirus Disease 2019 (COVID-19) Pandemic: A Qualitative Study

**DOI:** 10.3390/ijerph20105781

**Published:** 2023-05-10

**Authors:** Sonvanee Uansri, Watinee Kunpeuk, Sataporn Julchoo, Pigunkaew Sinam, Mathudara Phaiyarom, Rapeepong Suphanchaimat

**Affiliations:** 1International Health Policy Program, Ministry of Public Health, Nonthaburi 11000, Thailand; 2Division of Epidemiology, Department of Disease Control, Ministry of Public Health, Nonthaburi 11000, Thailand

**Keywords:** Coronavirus Disease 2019, health service, migrant, Thailand

## Abstract

The outbreak of Coronavirus Disease 2019 (COVID-19) has threatened health and well-being in all populations. This impact is also deepening structural inequalities for migrant workers in Thailand. Due to their vulnerability and limited opportunity to access health services, they have greater risks in many health aspects compared to other populations. This qualitative study sought to examine the key health concerns and barriers during the COVID-19 pandemic on healthcare access among migrant workers in Thailand through the lens of policymakers, healthcare professionals, experts on migrant health, and migrant workers. We conducted 17 semi-structured in-depth interviews of stakeholders from health and non-health sectors in Thailand from July to October 2021. The interviews were transcribed and analyzed using both deductive and inductive thematic approaches. Thematic coding was applied. The results showed that financial constraints were a major barrier for healthcare access among migrant workers. These included affordability of healthcare and difficulty accessing funds (migrant health insurance). Structural barriers included some health facilities opening for emergency cases only. Insufficient healthcare resources were profound during the peak of positive cases. Cognitive barriers included negative attitudes and diverse understanding of healthcare rights. Language and communication barriers, and a lack of information also played an important role. Conclusion, our study highlights healthcare access barriers to migrant workers in Thailand during the COVID-19 pandemic. Recommendations for future resolution of these barriers were also proposed.

## 1. Introduction

Coronavirus Disease 2019 (COVID-19) has been globally recognized as one of major public health threats in the past few years and was declared by the World Health Organization (WHO) on January 2020 as a public health emergency of international concern (PHEIC) [1,2]. The first case of COVID-19 was reported in Wuhan city of China; later the number of cases rapidly increased worldwide and became a global pandemic. According to a WHO report on 28 January 2022, the number of confirmed cases reached over 360 million with approximately 5.6 accumulated deaths.

Thailand was the first country that reported COVID-19 cases after China. During March–May 2020, the country faced a large cluster of infected related to superspreading events in boxing stadiums and nightlife hotspots [3,4]. Several social interventions and a whole-government approach with effective government actions were key contributory factors for the success in outbreak containment during the first wave [5]. However, there were large numbers of subsequent cases during the second wave of COVID-19. At that time, there was a huge influx of infective migrant workers who travelled illegally along the border of the country. The second wave was generated from a big cluster in the shrimp market in Samut Sakhon (a vicinity province of Bangkok) during December 2020–February 2021. Compared with the first wave, the number of cases was far greater in the second wave [6]. A third wave began in late March 2021. As of 1 April 2021, Thailand had recorded over 2 million cases and 20,000 deaths due to the third wave. The fourth wave of COVID-19 in Thailand peaked in mid-August 2022 due to the Delta variant.

Thailand has been known as a regional migrant hub in Southeast Asia due to its central location in the region. It shares a long border with four adjacent countries. The Thailand Migration Report in 2019 reported that there were approximately 4.9 million migrants in the country, mostly relocated from Cambodia, Lao PDR, Myanmar, and Vietnam (so called CLMV nations). Many migrants are undocumented and up to 49% of Thailand’s migrant population remains uninsured [7]. A recent study estimated that the volume of undocumented migrants equalled 800,000 (around 1/3 of total migrants).

Migrants’ vulnerability has been affected by many social determinants of health (such as housing, community, environment, education, and healthcare). Most migrants face difficulty in accessing health services due to reasons such as insufficient income, as well as cultural and language barriers which result in unsatisfactory and poor patient provider communication [8,9]. Cultural barriers in this regard also encompass mismatches between religious beliefs and lifestyles of migrants and the nature of health services in destination countries. COVID-19 has highlighted their situations and may make them more vulnerable. Crowded housing and unhygienic living conditions are found among most migrants and such situations partly facilitate rapid COVID-19 transmission [10].

Despite existing evidence showing that migrants in Thailand have been exposed to difficulties in accessing health services or health insurance [11,12], the systematic or scientific reports on barriers that migrants faced during COVID-19 are great in number. Therefore, this study aimed to explore the perceived barriers in accessing healthcare among migrant workers in Thailand, and the potential health impacts from those barriers during the COVID-19 pandemic in Thailand. We hoped that these findings would help inform the policymakers and relevant stakeholders to bring about policies to further promote migrants’ health and well-being.

## 2. Materials and Methods

### 2.1. Study Design

This qualitative study was carried out from July to October 2021 as part of a larger research project to study situations and recommendations for the health service system for migrant workers during the COVID-19 outbreak in Thailand.

This study adopted a model the “Healthcare Access Barriers (HCAB)” model suggested by Carrillo et al. to elaborate barriers to healthcare access in migrant workers. This model suggests interplay of financial, structural, and cognitive barriers, which cause inequalities in healthcare outcomes. It is a two-tiered model showing the connection between the findings with directional arrows. It demonstrates a top tier of intertwining barriers found to impede healthcare access (financial, cognitive, and structural as a barrier) and a bottom tier depicting the overall outcome of the previous factors (potential health outcome disparities) (Figure 1).

Semi-structured interviews were chosen to allow an in-depth exploration of their personal experiences, attitudes, and views to be gained. It also provided an appropriate format for discussing sensitive subjects whilst allowing a degree of flexibility to change questions to address areas important to each participant [14]. In this study, the focus was mainly on the supply side from the perspective of the providers. However, we still engaged a few migrants and staff of key non-government organizations (NGOs) which had extensive experience of working with migrant communities.

### 2.2. Sample and Data Collection

As previously mentioned, we included policymakers, healthcare professionals, experts on migrant health, and migrant health volunteers in the interview. The term “migrant health volunteer (MHV)” refers to “MHVs are assigned to be interpreters or health assistants in health facilities or NGOs and are required to collaborate and educate MHVs and migrant populations in particular areas. Publicizes health information, transfers knowledge, and provides guidance to other migrants on health matters such as health insurance registration, health-care eligibility, and duties of migrants in Thailand; notifies the public and reports to health officials about health information or abnormal situations and communicable diseases in the community; and serves as a health leader for migrant health development; encourages other migrants to join health development and service activities; and takes the lead in health behavior change in the migrant community” [15]. The final number and list of interviews are shown in Table 1. Twenty interviewees were included in the study (ten policymaker interviews, five healthcare professional interviews, two experts on migrant health interviews, and three migrant health volunteers). Snowball sampling was carried out if interviewees suggested further participants [16]. Due to travel restrictions during COVID-19, a phone interview was undertaken. The interview length was approximately an hour. Open-ended questions were used first, then we delved into specific topics, including: (i) interviewees’ experiences with migrant health policy; (ii) viewpoints towards national migrant policy in Thailand during COVID-19; (iii) the impact of the COVID-19 pandemic on health service system for migrant workers in Thailand; (iv) the status of healthcare access in informal/external migrant workers (financial, structural, and cognitive barriers); and (v) recommendations on health service systems for migration workers given any future outbreaks.

### 2.3. Analysis

Thematic analysis was used [17]. Interview transcripts were read in and analyzed. Then, an inductive approach was exercized to identify themes, categories, and new emerging codes and themes not previously identified within the HCAB mode. For confidentiality purposes, quotes are anonymized.

### 2.4. Ethics

This study followed the guidelines of the Declaration of Helsinki and was approved by the Institute for the Development of Human Research Protections in Thailand (IHRP 111/2564). At the point of recruitment, each participant was provided with an information sheet with details of the study. Signed consent was obtained for participation in the study, permission to be audio-recorded, and permission to be quoted anonymously in research outputs. The participants were allowed to refuse to answer in any of the interview questions at any point of time. The audio-recordings were anonymized prior to transcription to avoid disclosure of the participants’ identity.

## 3. Results

Barriers to healthcare access in the study area were categorized into three major themes: (1) financial barriers, (2) structural barriers, and (3) cognitive barriers. These themes are discussed separately but are interconnected as one barrier is a cause or effect of the other. Below, Table 2 summarizes themes and sub-themes that emerged from the research study.

### 3.1. Financial Barriers

According to the study findings, financial barriers proved to be a major barrier to healthcare access. Multifactorial components of this barrier exist and are a major determinant for ability to access healthcare.

Lack of migrant health insurance

Without insurance, there would be limited access to primary care and COVID-19 screening. Although in theory, all migrants should be insured, in reality, some migrants were not enrolled in public insurance (and remained undocumented). Therefore, for these undocumented migrants, secondary care referrals could not be made. Moreover, migrant workers were likely to be denied by healthcare professionals due to their uninsured status. This problem was even more complicated with the difficulty in the reimbursement process of the treatment cost (both in the normal situation and amidst the COVID-19 crisis). Therefore, health facilities needed to bear the cost of screening or treatment for undocumented migrants on their own.

“In the past, the government had a plan to take care of undocumented migrants who did not have any passports or any personal identity evidence…Yet, de facto, those migrants were not able to visit any health facilities… As such, this leads to refusal from health facilities to provide health screening for undocumented migrants especially in the COVID-19 pandemic”.(C01)

Unaffordability of hospital care

Unaffordability of hospital care cost also played an important role. Migrant workers were often paid daily wages. Thus, when they missed the work, they often lost their income. Some interviewees mentioned that migrant workers were not often given time off work to seek treatment, and some were penalized with fines for non-attendance. Many migrant workers had their employment summarily suspended or terminated during the pandemic. Some migrant employees left their jobs without pay, while others had their rates of pay reduced.

“With limited income, migrant workers or undocumented migrants could not afford to pay for medical bills. The bill is high”.(F10)

Unstable employment status

Employment status was shown to play a large role as unemployment is rife within the community and impacts upon the ability to acquire health insurance. All participants indicated that they had issues with the ability to pay for services or medicine regardless of their ability to access healthcare.

“COVID-19 interrupted our VISA extension, and then some migrant workers cannot continue their VISA validity. This resulted in negative impacts on access to health insurance. Moreover, some migrant workers needed to shift their work and then change their employers. This circumstance can limit our access to social security scheme and health insurance too. Also, the type of work can affect health screening, and it is even worse when they found that they were infected. Migrant workers in small companies needed to drop out of their work once they got infected, and it meaned that they lived without payment”.(C01)

“Once they have no jobs, it means that they lose their income”.(D01)

A rise in the number of positive cases in Thailand became a big challenge, especially in the first phase of the spread of COVID-19. The Thai government announced stricter measures to contain the pandemic. Workplace closure was implemented in factories or construction sites. However, there were no relief plans to mitigate the drawbacks of such measures. At the same time, it caused a burden of self-responsibility among employers and migrant workers themselves.

“The government is the key decisionmaker. Closure on the construction sites was implemented without any relief plans. We are unsure what we should do next”.(B01)

### 3.2. Structural Barriers

Constraint in health system design

During the peak of the confirmed cases, hospital beds were not available for all patients in need. Home isolation then became an alternative care choice for asymptomatic COVID-19 patients. However, it was almost impossible for most migrant workers, especially those in the construction sector, as they shifted their workplace and domicile from time to time and were mostly living in congested areas. Moreover, some migrant workers, especially the undocumented ones, feared of harassment by enforcement agencies. Thus, they were reluctant to visit official health facilities to seek treatment. In other words, the design of the health system, particularly during COVID-19, appeared to be incompatible with the working life and financial status of migrants, especially the undocumented. Moreover, the financial implications of healthcare placed a greater burden on undocumented migrants due to their limited access to the national health system, exacerbating existing disparities in healthcare access and outcomes.

“Home isolation for migrant workers is challenging. At the construction sites, their living conditions did not enable them to do self-quarantine. Some of them are living in small shelters and work transfer is quite common for them. In normal situation, poor living conditions are obvious, and it is even worse during this pandemic”.(F10)

Service adjustment during the COVID-19 period

There were several service challenges during the surge of COVID-19 cases in Thailand. Many hospitals postponed the appointments. Some health facilities opened services for emergency cases only. Some respondents mentioned that some hospitals prioritized the service by paying more attention to Thais over migrants. The situation was more complicated in undocumented migrants or migrants in detention centres where the government budget was sparse and was not clearly earmarked for this population. Moreover, the service system adjustment during COVID-19 appeared to overlook this group of the population.

“Over the first wave of the crisis, health facilities denied migrants with positive cases. Thai patients were priority. They urged us (migrants) to stay home”.(D01)

### 3.3. Cognitive Barriers

Negative attitudes towards migrant workers

During the rise of COVID-19 cases, most participants reported that there were negative attitudes surrounding migrants. The attitudes towards the idea that migrant workers were free-riders of the system and were the leading cause of COVID-19 spreading in certain areas.

“There were growing trends of negative attitudes towards migrant workers, particularly when new migrant COVID-19 cases were found in communities. Burmese migrant workers were prohibited from entering (some) fresh markets. There was a sign stopping them at the entrance. They were afraid that these people would bring more infection in the area. Surely, those migrants would face more difficulties in their daily life”.(F10)

Language and communication barriers

Language and communication barriers caused some migrants to have a lack of information relating to healthcare and how to adjust themselves in response to COVID-19. Some participants mentioned that migrants lacked information about the availability of services in their residential area. Information was often not acquired from official sources but was mostly acquired from informal networks such as neighbors, colleagues and family members, and migrant health volunteers. The inability to communicate frustrated doctors and sometimes this resulted in inadequate treatment. Language barriers also caused a risk of medication errors and poorly obtained consent for procedures. In addition, these barriers were linked with poor health literacy. Some migrants did not know how to protect themselves against COVID-19 and had little knowledge on the use of personal protective equipment.

”Yesterday we were informed that workplaces planned to contain the spread of COVID-19 by sealing the workplaces. This measure needed to be explained explicitly to migrant workers, and we needed to educate them…But the staff did not know how to explain them as we spoke in different languages…Some migrant workers could not access knowledge sources and had poor health literacy to take care of themselves during the spread of COVID-19. There was misunderstanding (in the sense of improper use) about self-protection; how to wear masks, hand gels, or soaps”.(C02)

## 4. Discussion

Migrant workers are vulnerable and are disproportionately affected by the impact of COVID-19. Migrants are being disproportionately affected by COVID-19 in most countries where data are available. Prior studies have shown that in Canada, Denmark, France, Germany, Italy, the Netherlands, Norway, Portugal, Sweden, the United Kingdom, and the United States, immigrants have a higher likelihood than native-born individuals to contract the virus, experience severe symptoms, and face a higher risk of mortality [18,19]. Such vulnerable conditions are shaped by poor living conditions, limited access to healthcare, limited knowledge, and poor health literacy. This problematic situation is coupled with the fact that health system adjustment during the surge of COVID-19 cases (though necessary) did not match the way of life of most migrants.

Based on the findings above, we found that the financial barrier does not act alone. It deeply intermingles with the precarious employment status and citizenship status of migrant workers. The precarious citizen status of migrants is recognized as a prime reason for the inability to access healthcare services as mentioned earlier in previous studies [20,21]. It is even more pronounced in the time of crisis, such as COVID-19, when migrants were more susceptible than general host populations. As Thailand curtailed economic activity and began to close its borders to limit the spread of COVID-19, some migrants lost their jobs during COVID-19 (and later dropped out of the social welfare system), and with limited household assets from the beginning (compared with the host population), the situation worsened. Most migrants thus heavily relied on ad hoc social assistance or support from NGOs or charitable groups [22]. A recent study in 2021 by the International Labor Organization (ILO) demonstrated that a number of migrant workers could not access at least one type of healthcare services [23]. The majority of them mentioned that it was because they could not afford healthcare, and they were likely to be forced to the healthcare payment [23]. From another angle, the COVID-19 crisis created opportunities for a country to strive for a more inclusive social protection for everybody (including migrant populations) on its soil. The idea of how to expand health coverage to undocumented migrants in the informal sector, and to enhance their access to the formal welfare system needs to be discussed [12]. To expand coverage to undocumented migrants, support from the government is needed alongside a strong collaboration with civil societies. This idea is not new, as at the international level, the Thai government has already ratified the Global Compact for Safe, Orderly and Regular Migration (GCM), which emphasizes the importance of an inclusive society as part of the health security of a nation [24]. Moreover, the spread of COVID-19 can be an opportunity to rethink about the integrative information system for undocumented migrant registration, which requires coordination across the government ministries. With this idea, there is substantial room to use COVID-19 as a jumping platform for many heath initiatives; for instance, the use of digital platforms and technology connectivity to enhance the enrollment of vulnerable populations (especially migrants) into the service system and to promote health literacy amongst migrants.

We also noticed the difficulty in accessing healthcare information amongst migrants. A prior cross-sectional study on migrants’ risk communication in Thailand exhibited that middle-aged migrants seemed to have a lower degree of awareness on prevention measures than the younger ones [25]. Although there are booklets, leaflets, and online platforms to combat xenophobia and promote social cohesion during the crisis [26], the negative attitude of the society on migrants still remains.

The unemployment status of migrant workers during the crisis was also mentioned in our finding. This is consistent with the findings from ILO which showed that migrant workers tended to lose their jobs as employers were likely to reduce the number of the workforce [27]. It was estimated that about 700,000 migrant workers in tourism, services, and construction industries were likely to be unemployed since the beginning of the country lockdown in March 2020 [27]. Moreover, a cross-sectional study conducted by ILO among 2187 low-skilled migrant workers in 2021 revealed that about 42% of respondents were paid lower than the national minimum wage [23], although prior to the onset of COVID-19 about one third were paid lower than that number. The evidence also suggested that more than half of migrant women were paid lower than the minimum wage [23]. For the unemployment rate, about a quarter of respondents reported that they were not employed during the lockdown from May to June 2020, and women had more potential to be unemployed over men [23]. There were actions by the Thai government to simplify regularization and extend the visa of migrant workers [28]. Yet, the challenge is how to overcome bureaucratic difficulties on the ground.

Inequality of healthcare access between the Thais and migrant workers became debatable both before and during the COVID-19 pandemic. Structural challenges in the healthcare system were profound and these gaps remained critical over this crisis. One of the key challenges was how to implement the COVID-19 measures while adapting them in a way that matched the lifestyle of migrants. Poor living conditions of migrant workers were one of the key barriers. Although everybody admits that self-quarantine is a key measure to prevent further spreading of the disease, this cannot be done easily in the households or workplaces of migrants. Evidence has demonstrated that in most workplace settings, the majority of migrant workers could not conform with social distancing measures [23]. Some lived in congested dormitories where physical distancing was nearly impossible.

However, the most challenging point in moving toward an inclusive society is how to shape the attitude and mindset of the government and society to recognize the benefit of having an inclusive society where migrants are not left behind rather than leaving them undocumented (and then creating hidden service burden to the health system) [22]. According to the findings above, we also found that negative perceptions toward migrants persisted before the crisis and even deepened during the crisis. An obvious example is the belief in the society that the second wave of the epidemic in Thailand originated from migrant workers in Samut Sakhon. To overcome this issue, social empowerment is key (alongside seamless collaboration with the government). Further studies demonstrating the advantage of including migrants in the service system at the beginning (rather than overlooking them) are of great value.

Some limitations remained in this study. First, as most participants were recruited by purposive sampling and snowball sampling, they could not be well representative of the study population (although transferability of the results to the same kind of participants is possible). However, since the nature of qualitative research is to maximize variability of the participants and identify the hidden constructs from the interviewees’ view, we do not think statistical representativeness is the main concern (although the authors should be aware of the finding interpretation). Second, the viewpoints of the interviewees could change over time as the pandemic progressed. Third, some interviews were conducted by a team member who was from an authority in the Ministry. This can be both a strength (that the interviewer tended to understand the setting context) and a limitation (where some interviewees might tend to express their views that favor the Ministry’s operation). We tried to minimize this bias by triangulating the interview findings of each interviewee with one another. Future studies that explore the health needs of subgroups of migrants and further explore the adaptation of the system and migrant communities themselves after the pandemic are recommended.

## 5. Conclusions and Recommendations

All in all, this study shed light on the barriers of healthcare access faced by migrants. These barriers included, but were not limited to, the precarious status of migrants, the lack of health insurance, the loss of job during the pandemic which later caused them financial difficulty, and the health service reorientation in response to the pandemic that did not match the way of life of some migrants. The aforementioned problems were also coupled with negative attitudes towards migrants in society. With all accounts above, the Thai government should consider the following policy recommendations. First, expanding health insurance coverage to existing undocumented migrants is needed. This also means the reorientation of the characteristics of the insurance scheme to be more attractive and more affordable. Second, there should be social safety for people in the time of crisis; for instance, a guarantee for basic health needs for those losing jobs. This idea should be applied to all people in society. Third, initiatives to overcome language and cultural barriers are crucial. The government should provide financial support and resources for public facilities in migrant-dense areas to have professional interpreters and bilingual materials for health information. Finally, social communication is needed to change the mindset of the society towards social inclusiveness and recognizing people from diverse origins as part of the entire society. This requires seamless cooperation from all partners, including the media, the health sector, the education sector, the business sector, academic, and civic groups.

## Figures and Tables

**Figure 1 ijerph-20-05781-f001:**
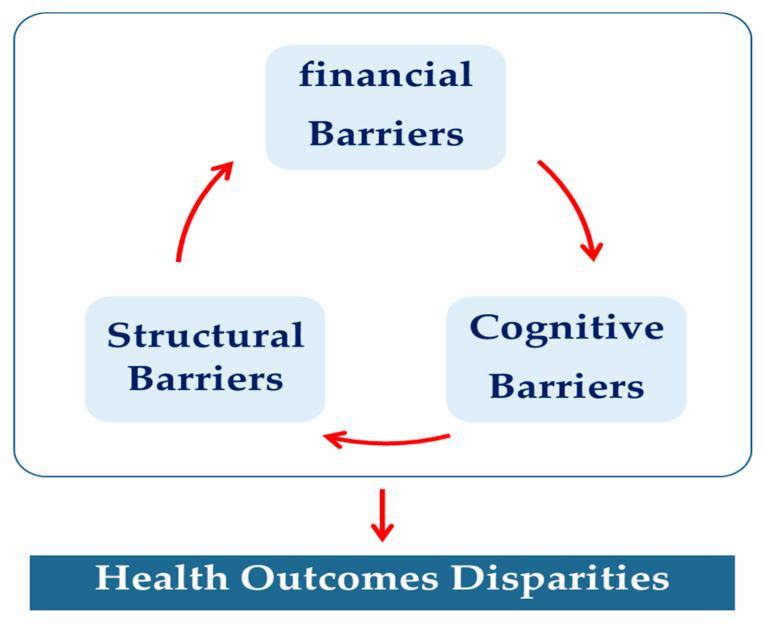
Healthcare access barriers (HCAB) model suggested by Carrillo et al. (2011) [13].

**Table 1 ijerph-20-05781-t001:** List of interviewees about barriers to migrant workers healthcare access.

Code	Involvement with Social and Health Issues among Migrant Workers
Policymaker (Ministry of Public Health)	
A01	Consultant of Deputy Permanent Secretary, Ministry of Public Health
A02	Head of Health Administration Division, Ministry of Public Health
A03	Consultant of Department of Medical Services, Ministry of Public Health
A04	Policymaker from the Division of Health Economics and Health Security, NHSO
A05	Policymaker from the National Institute of Emergency Medicine
Policymaker (Other Ministry Participants)	
A06	Advisor to the Minister of Social Development and Human Security, Thailand
A07	Deputy Permanent Secretary of National Health Commission Office, Thailand
A08	Head of National Health Security Office
A09	Head of health promotion plan for vulnerable populations, ThaiHealth Promotion Foundation, Thailand
A10	Staff of health promotion plan for vulnerable populations, ThaiHealth Promotion Foundation, Thailand
Healthcare professionals	
B01	Medical doctor from Department of Disease Control, Ministry of Public Health
B02	Medical doctor from Department of Disease Control, Ministry of Public Health
B03	Public health academic who provides healthcare for detainees (Immigration Bureau)
B04	Nurse who provides healthcare to detainee
B05	Volunteer medical doctor who provides healthcare for COVID-19 patients in home isolation
Experts on migrant health (non-government organisations: NGOs)	
C01	Staff who have experience in work related to health services and quality of life of migrants and/or foreigners
C02	Staff who have experience in work related to health services and quality of life of migrants and/or foreigners
Migration workers	
D01	Migrant health volunteer
D02	Migrant health volunteer
D03	Migrant health volunteer

**Table 2 ijerph-20-05781-t002:** Highlights the themes and subthemes from the interview.

Theme	Subthemes
Financial Barriers	1. Lack of migrant health insurance
	2. Unaffordability of hospital care
	3. Unstable employment status
Structural Barriers	1. Constraint in health system design
	2. Service adjustment during COVID-19 period
Cognitive Barriers	1. Negative attitudes towards migrant workers
	2. Language and communication barriers

## Data Availability

Data available on request due to ethical restrictions.

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
