# Peer review of "Perceived Barriers of Accessing Healthcare among Migrant Workers in Thailand during the Coronavirus Disease 2019 (COVID-19) Pandemic: A Qualitative Study"

_ijerph, 2023, doi:10.3390/ijerph20105781_

Round 1
Reviewer 1 Report (Previous Reviewer 2)
The authors of this manuscript have adequately addressed the concerns presented by the reviewers.
Author Response
Thank you very much.
Reviewer 2 Report (Previous Reviewer 1)
I thank the authors for the review they have done. I think it is a good and valuable study, and it offers useful insights on the subject matter. Certain imbalances in the text remain still. I would recommend the authors to read the manuscript once again with lenses of objectivity and comparative knowledge for other regions in the world, to ensure a more balanced narrative. Some concrete suggestions are below; similar observations may be expanded for the parts of the narrative. I would also recommend proof-reading the draft for the language. Please also be careful and accurate with the references and citations. Some of the citations do not confirm the statements in the body of the article, or confirm marginally.
14: “threatened impacted” – one of the two words should be deleted?
65: “Many migrants are undocumented” – “many” is a value statement. Please add percentages (to the total migrant cohort) or reformulate.
67: “due to many reasons, such as insufficient” – same as above, can be “due to such reasons as..”
66-67: “Most migrants face difficulty in accessing health services due to many reasons, such as insufficient income, cultural and language barriers” – please clarify what is means by cultural barriers and how they hinder access to health services. Same comment to line 374.
68: “Although migrants’ vulnerability has been known” – this needs to be explained. The sentence below may be moved up and reformulated for this purpose.
212: “In other words, the design of the health system, particularly during the COVID-19, appeared to be incompatible with the lifestyles of migrants” –
1) all migrants or undocumented migrants?
2)the word “lifestyles” imply the existence of a choice for the migrants, which does not seem to be the message of the paragraph. Perhaps specify that the financial implications of the treatment placed a greater burden on the (un)documented due to the lack of access to the national health system (if this is what the message should be, please kindly reformulate to suit your idea).
222-223: these were not specific to migrants but generic for all categories of population, and not only in Thailand but world-wide.
Section 3.3. interchangeably uses discourses and attitudes. When using the word “discourses” in an evaluative sense, one would need to offer a discourse analysis result, to substantiate the claims.
246-250: “lack of information” and “having little knowledge” are different things.
This entire section suggests some kind of intent at information restriction, which is unlikely.
260-162: “There was misunderstanding about self-protection; how to wear masks, hand gels, or soaps.” – I understand that this a quote but what is meant by a misunderstanding about how to wear masks/use soap or hand gels?
265-268: “Immigrants were much more likely than native-born peers to contract the disease, to experience severe symptoms, and to have a higher mortality risk in almost all of the countries for which data are available” – does the source state this actually? For Italy, it mentioned that a significant part of the foreign population consists of young migrants and native-born children of immigrants, who are less likely to show COVID-19 symptoms. Higher mortality risks also seem to be connected to the economic factors for the migrant populations. Is there a comparative study focusing on the comparison of COVID effects amongst natives and migrants in a similar economic situation in the same geographical area?
5. Conclusions and recommendations section is useful. The title is “Perceived barriers of accessing healthcare among migrant worker in Thailand during the Coronavirus Disease 2019 (COVID-19) pandemic: A qualitative study” – and recommendations have suggestions to The Thai government, while in reality the section reads as a whole-of-society approach. This is also an opportunity to call for greater agency on the part of migrants themselves, to pro-actively seek information and follow the rules. Conclusions should be in line with the titles perhaps?
Author Response
Dear editors and reviewers
Thank you so much for the comments on the manuscript. We do appreciate your help in making this manuscript to have a better shape. We have addressed the comments on a point-by-point basis. Please find our responses below. The marked-up manuscript and the clean manuscript are re-submitted in parallel with this letter.
Best regards
All authors

Round 2
Reviewer 2 Report (Previous Reviewer 1)
I would like to thank the authors for addressing the comments and editing the narrative. It reads better. It is truly a pity that the publication is only going out now; its relevance would have been greater earlier (2022).
One minor suggestion is to update the WHO numbers on cases of COVID, which currently is for Jan 2022. If data are available, I would advise to include more recent numbers, given that the publication is going out in April 2023.
This manuscript is a resubmission of an earlier submission. The following is a list of the peer review reports and author responses from that submission.
Round 1
Reviewer 1 Report
I would like to sincerely thank the authors for drawing the attention to the matter of migrant health and access to health services in Thailand, with a specific focus on the features that surfaced during the COVID pandemic. Given the estimates of migrant workers in the country, in total and in proportion to the population, the proposed topic seems relevant and calling for both research and policy responses. The difficulties experienced by the group represent an additional relevant angle to the study. It is regrettable that the article only is submitted for a publication now, when the topic of the impact of the pandemic on (undocumented) migrant workers has been explored in much detail. Clearly, Thailand represents a case study in this context, but the findings largely confirm the findings for the other countries. From this point of view, its originality is marginal – unless the authors would like to strengthen this section. For a publication, I would suggest to add a stronger justification for the study, with statistical references/ data as to why this text merits a read.
It would be good to see a clearly formulated research question.
The draft would benefit from proofreading and correction of typos.
Below, I propose a few specific suggestions to address, which would improve the readability and the flow of the text.
Absract generally: is it referring to Thailand, or? There is a range of broad statements, that may or may not be accurate for different locations. It would benefit from precision and concretisation.
13: “potentially”: the pandemic is a matter of the past, so has it or has it not?
97: “migration workers” should be defined. 111 defines tem as migrant health volunteers, should this be used throughout the text? also, it would be helpful to define who migrant health volunteers are – are these doctors? Nurses? Neither? How are they identified? By the scholars? Self-identified?
133: how are “Negative discourse on migrants” and “cognitive barriers” related in this table?
162: “high” is a value statement. Should this sentence be evidenced by average wage information and average bill information?
164: within should be written together. There are multiple spelling errors in the text, which are easy to detect and to correct – recommended.
194: “it” refers to?
210-211: a recommendation to stay at home was/remains a standard response to the reported COVID cases until the present day, unless heavy medical intervention/treatment is necessary. This was/is universal response and is not specific to migrant workers.
212-213: was this the case in other countries? Where did this exist, which states introduced this function? Please add.
215: discourse does not seem to be a suitable word here. a sign is not equivalent to a discourse, and the article does not focus on discourse analysis on migration or migrant workers.
245: are there data confirming this?
256: should it be “from”?
256-257: the meaning is not clear.
340-341: which data confirms this statement? This is not part of the research question/focus, either.
345-346: does this apply to the Thai population?
Reviewer 2 Report
Line 37 please denote the epicenter of the COVID-19 virus was Wuhan, China not just the country.
LLine 79 has an APA 7th edition citation error et al. should have a year of after the author
Line 100 you need to cite a source regarding the snowball sampling method
Line 113 you need to cite a source regarding thematic analysis in qualitative methodology.
